# Drivers and Barriers Influencing Adherence to the Mediterranean Diet: A Comparative Study across Five Countries

**DOI:** 10.3390/nu16152405

**Published:** 2024-07-25

**Authors:** Chiara Biggi, Beatrice Biasini, Nives Ogrinc, Lidija Strojnik, Isabella Endrizzi, Leonardo Menghi, Ikram Khémiri, Amani Mankai, Fethi Ben Slama, Henda Jamoussi, Katerina Riviou, Kaoutar Elfazazi, Nayyer Rehman, Francesca Scazzina, Davide Menozzi

**Affiliations:** 1Department of Food and Drug, University of Parma, Parco Area delle Scienze, 27/A, 43124 Parma, Italy; chiara.biggi@unipr.it (C.B.); beatrice.biasini@unipr.it (B.B.); francesca.scazzina@unipr.it (F.S.); 2Jožef Stefan Institute (JSI), Jamova 39, 1000 Ljubljana, Slovenia; nives.ogrinc@ijs.si (N.O.); lidija.strojnik@ijs.si (L.S.); 3Research and Innovation Center, Edmund Mach Foundation, Via Edmund Mach 1, 38098 San Michele all’Adige, Italy; isabella.endrizzi@fmach.it; 4Center Agriculture Food Environment, University of Trento, Via Edmund Mach 1, 38098 San Michele all’Adige, Italy; leonardo.menghi@unitn.it; 5Tunisian Association of Nutritional Sciences, National Institute of Nutrition and Food Technology, 11 Rue Jebel Lakhdar, Bab Saadoun, Tunis 1007, Tunisia; ikram.khemiri@fst.utm.tn (I.K.); amani.mankai@yahoo.fr (A.M.); benslamafethi@yahoo.fr (F.B.S.); henda.jamoussi@fmt.utm.tn (H.J.); 6Laboratory of Mycology, Pathologies and Biomarkers (LR16/ES05), Faculty of Sciences of Tunis, University of Tunis El Manar, Campus Universitaire, Tunis 2092, Tunisia; 7Higher School of Health Sciences and Techniques, University of Tunis El Manar, B.P. 176–Bab Souika, Tunis 1007, Tunisia; 8Research Unit “Obesity: Etiopathology and Treatment, UR18ES01”, Faculty of Medicine, University of Tunis El Manar, 15 Rue Jebel Lakhdar, Bab Saadoun, Tunis 1007, Tunisia; 9Ellinogermaniki Agogi Scholi Panagea Savva AE (EA), Dimitriou Panagea Str., 15351 Pallini, Greece; kriviou@ea.gr; 10Agri-Food Technology and Quality Laboratory, Regional Center of Agricultural Research of Tadla, National Institute of Agricultural Research, Morocco (INRA), Avenue de la Victoire, B.P. 415 RP, Rabat 10090, Morocco; kaoutar.elfazazi@inra.ma; 11WRG Europe Ltd., 26-28 Southernhay East, Exeter EX1 1NS, UK; nayyer@wrgeurope.co.uk

**Keywords:** Mediterranean diet, attitude, behavior, picky eating, food choice motives, sustainability food choice motives

## Abstract

Given the global decline in adherence to the Mediterranean Diet (MD), even within its native region, it is key to identify the factors influencing this trend to mitigate the negative health outcomes associated with westernized diets. To this end, 4025 individuals (49.6% women, 42.6 ± 14.2 y/o) from Greece, Italy, Morocco, Slovenia, and Tunisia remotely completed a series of measures assessing motives, attitudes, and psychosocial factors related to MD adherence, which was evaluated using the MEDAS questionnaire. The results suggested medium-to-low adherence across all countries, with the highest adherence in Italy and Morocco and the lowest in Slovenia. Structural equation modeling revealed that positive attitudes toward the healthiness of food were the strongest predictors of adherence, whereas picky eating was a significant negative predictor in all countries except Greece. Adherence to the MD was positively influenced by health motivations in Morocco and weight control in Slovenia and Greece, while sensory appeal negatively influenced adherence in Italy. Additionally, price and convenience were significant barriers in Tunisia and Greece, whereas a preference for local and seasonal foods promoted adherence in Morocco and Greece. Overall, our findings underscore the need for country-specific interventions and policies that address distinct local factors and motivations to ease favorable shifts in dietary patterns toward MD principles.

## 1. Introduction

The Mediterranean Diet (MD) is deeply rooted in the cultural and agricultural traditions of countries around the Mediterranean Sea [1]. The MD was first described by Ancel Keys in the 1960s, based on eating habits in Greece and Southern Italy. This diet is known for being low in saturated fats and high in vegetable oils. It includes plenty of vegetables, fruits, nuts, seeds, legumes, whole grains, and olive oil, with moderate amounts of fish, poultry and alcohol (mainly wine), and limited dairy, red meat, processed meats, and sweets [2,3].

It has long been followed by these populations for its emphasis on the quality and origin of food, using fresh, locally sourced, and seasonal ingredients. Not only does the MD improve physical health but it also promotes social interactions through communal meals, enhancing social and mental well-being [4,5]. The positive social impact and the improvement in healthiness due to adherence to the Mediterranean Diet (MD) are supported by several epidemiological studies [6]. The MD has been linked to many health benefits, particularly its ability to reduce the risk of heart disease [7]. It improves cardiovascular health by lowering LDL cholesterol and reducing inflammation, as well as decreasing the risk of stroke [8,9,10]. The MD has also been found to positively impact reproductive health and metabolic health [11], diabetes management [12], and certain cancers [13]. Beyond physical health, it potentially lowers the risk of cognitive dysfunction and neurodegenerative disorders, particularly Alzheimer’s disease [5].

Moreover, the MD has a relatively low environmental footprint concerning water, nitrogen, and carbon due to its focus on mainly plant-based options and legumes. When the Mediterranean diet is embraced not only for its health benefits but also as an environmentally sustainable dietary choice, it can positively contribute to biodiversity preservation and foster local economic sustainability [14,15].

Despite its well-known benefits, the adoption of the MD has been gradually decreasing [16], as well as in several countries around the Mediterranean Basin [17,18,19]. In fact, Kamiński et al. [20] found that the MD ranked 16th in their global diet popularity study. These findings may tentatively be attributed to the indirect consequences of social and cultural changes, increasing urbanization [21], as well as to the globalization of food systems within Mediterranean countries [22,23,24]. People have shifted toward a more western type of diet, with an associated reduced consumption of fruits and vegetables [25], in favor of more convenient, quick to prepare, and processed alternatives.

One of the main causes for this could be attributed to limited awareness and understanding of the health benefits of the MD [26]. Understanding consumer behavior is essential because attitudes, preferences, and awareness directly influence dietary choices. Often, cultural habits and personal food preferences make it challenging for people to adopt a new dietary pattern, even when they are aware of its benefits. The discrepancy between consumers’ attitude and awareness of healthy eating may act as a barrier to its adoption, potentially shifting consumers’ habits away from MD regime [27]. By addressing consumer needs and preferences, strategies can be developed to enhance the appeal and accessibility of the MD.

Given this context, there is a growing need for new studies to elucidate the main predictors of the decline in the MD adherence within its native regions. This need is further underscored by a recent systematic review on drivers and barriers to adherence to the MD [28], in which only ~22% of the studies (4 out of 18) included in the assessment were conducted in Mediterranean countries, namely in Europe (i.e., Greece, Italy, and Spain). Another literature review [14] has shown the need for large-scale studies of MD adherence in African Mediterranean countries, as they were underrepresented in the current literature. In particular, the study indicates the need for understanding the main determinants of adherence to the MD in these countries to help in designing appropriate local and national policies and interventions to promote it [14]. Indeed, given the substantial cultural and socioeconomic diversity of countries in the Mediterranean basin, it is evident that well-known mediators or barriers affecting MD adherence (e.g., sustainability beliefs, food choice motives, and attitudes toward healthy eating) may operate differently according to the context.

The present study applied quantitative research methods to investigate the main drivers and barriers affecting the adoption of healthy eating habits in five Mediterranean countries (Greece, Italy, Morocco, Slovenia, and Tunisia). A survey was performed to identify consumers’ motivations, attitudes, drivers, and barriers toward dietary patterns inspired by the MD, including the role of picky eating (PE), attitude toward healthiness of food, and local determinants of food choices. An online questionnaire was administered to a sample of 4025 individuals (approximately 800 subjects for each country) in their native languages (i.e., Greek, Italian, Arab/French, and Slovenian).

The identification of barriers and motivators influencing the adoption of healthy eating habits and their role in enhancing the discrepancy between consumers’ attitude and awareness in addition to the suggestion of effective strategies to foster healthy eating habits are among the main purposes of the present study.

## 2. Theoretical Framework and Research Hypothesis

A preliminary qualitative phase, developed with three focus groups in each of the five countries, allowed us to identify the main drivers and barriers able to influence the adoption of healthy eating and MD pattern [29].

Potential motives for choosing food were derived from the studies applying the Food Choice Questionnaire (FCQ), as described by Steptoe et al. [30], as well as the adapted version developed by Pieniak et al. [31] for the case of traditional food products. The selection of items was based on findings from the exploratory focus group discussions [29] and from the literature review. In our case, seven dimensions were included, namely weight control, price, convenience, natural content, health, sensory appeal, and familiarity. Health, weight, and natural content have been found to be positively related with vegetable consumption [32]. As extensively discussed in Section 1, the MD is also associated with several health benefits; the natural content was found to be a significant driver to the MD in a Portuguese immigrant community in the US [33]. Therefore, our hypotheses for these dimensions were as follows:

**Hypothesis** **1.**
*Those giving higher importance to the health factor have a higher adherence to the Mediterranean Diet.*


**Hypothesis** **2.**
*Those giving higher importance to the natural content factor have a higher adherence to the Mediterranean Diet.*


**Hypothesis** **3.**
*Those giving higher importance to the weight control factor have a higher adherence to the Mediterranean Diet.*


In different studies carried out in the US, convenience and sensory appeal were observed to be significant barriers to the MD [33,34]. Convenience, familiarity, and price have been identified as relevant barriers for the adoption of sustainable dietary behaviors in different Mediterranean countries, like France [35] and Italy [36]. Therefore, we considered the following hypotheses:

**Hypothesis** **4.**
*Those giving higher importance to the sensory appeal factor have a lower adherence to the Mediterranean Diet.*


**Hypothesis** **5.**
*Those giving higher importance to the convenience factor have a lower adherence to the Mediterranean Diet.*


**Hypothesis** **6.**
*Those giving higher importance to the price factor have a lower adherence to the Mediterranean Diet.*


**Hypothesis** **7.**
*Those giving higher importance to the familiarity factor have a lower adherence to the Mediterranean Diet.*


Several studies have highlighted the relationship between adherence to the MD and sustainable dietary behaviors [37]. The ethical dimension, given the importance of the environmental and social sustainability connected with the MD, was assessed following the short version of the Sustainable Food Choice Questionnaire (SUS-FCQ) developed by Verain et al. [38]. The SUS-FCQ distinguishes general sustainability, covering environmental, ethical, and animal welfare aspects as well as local and seasonal motives. Hence, Hypotheses 8–9 are as follows:

**Hypothesis** **8.**
*Those giving higher importance to the general sustainability factor have a higher adherence to the Mediterranean Diet.*


**Hypothesis** **9.**
*Those giving higher importance to the local and seasonality factors have a higher adherence to the Mediterranean Diet.*


Thus, attitude toward the behavior is a precursor of the intention to perform the behavior and of the behavior itself [39]. In our case, we investigated the effect of a general interest in eating healthily [40] and the effect of attitude toward the adoption of the MD [41] on the MD adherence and we suggested the following:

**Hypothesis** **10.**
*Those with a more favorable attitude toward the Mediterranean Diet have a higher adherence to the Mediterranean Diet.*


**Hypothesis** **11.**
*Those with a more favorable attitude toward healthy eating have a higher adherence to the Mediterranean Diet.*


Finally, the analysis examined the predictive value of picky eating as a selective eating behavior that may hinder adherence to the MD. Picky eating refers to unwillingness to include a wide range of both familiar and unfamiliar foods in the diet [42] and was confirmed to be an important barrier to dietary variety and quality. Several studies, indeed, evidenced a close negative link between picky eating and habitually consumed fruit and vegetables or overall dietary variety [43,44]. Moreover, Menghi et al. [42] reported evidence negatively linking adult picky eating to adherence to the MD within Italian adults. Hence, our hypothesis was as follows:

**Hypothesis** **12.**
*Adult picky eating is negatively associated with adherence to the Mediterranean Diet.*


The model was tested in the five countries, followed a similar pattern to the one developed by Pieniak et al. [31] for traditional food products, also allowing us to test the association of the motives for food choice with both adherence to the Mediterranean Diet and attitude toward the Mediterranean Diet (Figure 1).

## 3. Materials and Methods

### 3.1. Data Collection and Sample

The present study was approved by the local institutional Ethical Committee (REB—Research Ethics Board, 28-2023-N, 29 March 2023) and conducted according to the ethical principles stated in the Declaration of Helsinki. Data collection was conducted in all of the five countries (Italy, Greece, Morocco, Slovenia, and Tunisia) during the period October–November 2023, by an external agency. Representative samples were drawn from the national adult populations (aged 18–79) across age, sex, and geographical regions. To be part of the study, respondents were asked to electronically provide the informed consent, which was shown at the invitation page of the online survey. All participants were informed that no data could be identified or linked to individual persons and that data would be analyzed anonymously. The following screening out questions were established at the beginning of the survey: the survey was initially limited to those residing from at least one year in the country of residence for food habits adaptation. Then, we collected data from adults and aged people considering the age range provided by PubMed, thus excluding those younger than 18 years old and those 80 years old or older. Finally, we excluded those who were not at all responsible for food purchases for their family.

A quality check was performed on the final sample data to avoid the low-quality responses, following recommendations to assess and improve data quality in online questionnaires [45,46]. As a priori screening, an instructed response item was inserted in a long matrix (i.e., to demonstrate that you are not a robot, for this statement, please select “Strongly agree”). As a posteriori screening, subjects presenting straight-line patterns followed by fast responders were excluded from the final sample as they were deemed to be careless. Straight liners were identified based on a standard deviation equal to 0 calculated on the answers given to two question matrices (i.e., one referred to the attitude toward healthfulness of foods and one referred to drivers and barriers to the adoption of the Mediterranean diet) in each country. Indeed, both these two matrices included items with opposite meanings (i.e., “The healthiness of food has little impact on my food choices” and “I am very particular about the healthiness of food I eat”; “Mediterranean Diet contains lower-priced foods” and “Mediterranean Diet contains high-priced foods”), implying different answers by careful respondents and thus resulting in a standard deviation different to 0. Subsequently, fast responders were determined for each country as those who completed the survey in less than 40% of the median time calculated from participants who responded within a maximum of 60 min, similarly to what has been reported previously [47].

### 3.2. Measures

An online questionnaire was designed on a theory-driven approach. The questionnaire is provided in the Appendix A. All the items were selected either from previously validated measures within the countries involved (e.g., [42]), or have been adapted from previous studies on attitudes, intentions, and behaviors toward healthy eating (e.g., [40]) and the Mediterranean diet (e.g., [41]). For this latter scenario, the items were first translated from English into the native language by the partners and then back-translated from the native languages in English by a professional agency. Any discrepancies between the two English versions allowed for the identification of critical translations into the native languages, which were resolved by the partners to fully ensure the linguistic validity and consistency of the questionnaire. A first pilot administration test was performed in Italy, which confirmed the clarity and internal validity of the measures. Data collection was conducted using the platform Qualtrics (©2020 Qualtrics LLC, Seattle, WA, USA; Provo, UT, USA). The measures employed were as follows.

#### 3.2.1. The Food Choice Questionnaire

The food choice motives were investigated using the Food Choice Questionnaire (FCQ). The FCQ, developed by Steptoe et al. [30], was designed as an instrument to assess the relative importance of a range of factors related to dietary choice to individuals. We adapted the version by Pieniak et al. [31]. Twenty-one items were included, representing seven domains: health, convenience, sensory appeal, natural content, price, weight control, and familiarity. Items (e.g., “It is important to me that the food I eat on a typical day contains a lot of vitamins and minerals”) were measured on a 7-point scale (1—Not at all important; 7—Extremely important) and scores for each domain were computed by averaging the relative items, with higher values being indicative of higher importance associated with the domain.

#### 3.2.2. The Sustainable Food Choice Questionnaire

The Sustainable Food Choice Questionnaire (SUS-FCQ) was developed as an addition to the FCQ considering the full concept of sustainability [38]. Therefore, we assessed the sustainable food choice motives with ten items (e.g., “It is important to me that the food I eat on a typical day is produced without animals being in pain”) on a 7-point scale (1—Not at all important; 7—Extremely important) representative of four factors (i.e., animal welfare, ethical concerns, environmental welfare, and local and seasonal). The first three factors (i.e., animal welfare, ethical concerns, and environmental welfare) were averaged into a single score of “General sustainability”, following the approach of Verain et al. [38], with higher values indicative of higher importance associated with the general sustainability domain. The factor “animal welfare”, as well as the “general sustainability” score, was computed either with and without the item “Is produced in Halal way”, in order to consider this factor in the Muslim-majority countries.

#### 3.2.3. Health and Taste Attitude Scale

General interest in eating healthily was assessed using the “General Health Interest” subscale of the Health and Taste Attitude Scales (HTAS) by Roininen et al. [40]. This domain comprises eight items (e.g., “I always follow a healthy and balanced diet”, and “It is important for me that my diet is low in fat”), which were measured on a 7-point scale (1—Strongly disagree; 7—Strongly agree). Prior to computing a global score by averaging individual ratings, negatively keyed items (n = 4) were reversed.

#### 3.2.4. Adult Picky Eating Questionnaire

Picky eating was measured using the 20-item version of the Adult Picky Eating Questionnaire (APEQ) [42,48,49]. Adult picky eating measures unwillingness to eat unfamiliar foods or try novel foods and the APEQ is organized into four domains: meal presentation, food variety, meal disengagement, and taste aversion, which comprehensively assess various facets of picky eating behaviors (e.g., ‘I have a strong preference for specific food presentation’, ‘I eat a limited number of items from each food group’). Respondents rated each item on a 5-point scale (1—Never; 2—Rarely; 3—Sometimes; 4—Often; and 5—Always). A global score was computed by averaging the 20 items, with higher values indicating a higher level of picky eating.

#### 3.2.5. Assessment of Attitudes, Adherence, Drivers, and Barriers to the Mediterranean Diet

We assessed the direct measure of attitude toward the adoption of the Mediterranean diet with six semantic differentials, using a 7-point bipolar scale, e.g., “Following the Mediterranean Diet for me would be: Disgusting–Tasting” [41].

Adherence to the MD was assessed through the 14-item MEDAS questionnaire [50]. The objective of this section, which consists of 14 items (e.g., “How many teaspoons of olive oil do you consume in a given day (including that used for frying, salads, out-of-house meals, etc.)?”), was to understand how closely the population adheres to the recommendations of the MD pattern regarding daily or weekly consumption of typical MD foods. The frequencies of consumption were assessed for the following items: olive oil, fruit, vegetables, pasta and grains, legumes, fish, white meat, eggs, milk and dairy products, sugary beverages, red meat, and sweets. Each question was scored 0 or 1 (e.g., one point was given for using 4 or more tablespoons of olive oil/day). According to the MEDAS screener responses, a score was calculated, ranging from 0 to 14. Higher scores indicate higher adherence to the MD [50].

We then asked, on a 7-point scale, to what extent the respondents agreed or not (1 = Strongly disagree–7 = Strongly agree) to a set of items listing the possible drivers and barriers to the adoption of the Mediterranean diet. Each domain was assessed using between one and four items. When multiple-items were used, a single score for the driver/barrier was computed by averaging the relative items, with higher values indicative of higher importance associated. Among the drivers, factors such as health (e.g., “Mediterranean Diet has a positive effect on cholesterol”), diet quality (e.g., “Mediterranean Diet includes healthier and more nutritious foods”), applicability (e.g., “Mediterranean Diet is tastier and more sustainable than other types of diets”), lifestyle (e.g., “Mediterranean Diet increases consumption of homemade foods”), affordability (e.g., “Food access is easier in Mediterranean Diet”), and the environment lifestyle (e.g., “Mediterranean Diet has a positive effect on the environment”), were investigated. The barriers investigated were health (e.g., “Mediterranean Diet contains more allergenic foods”), lifestyle (e.g., “Following Mediterranean Diet is difficult due to conflict with cultural habits/beliefs/norms”), and affordability (e.g., “Mediterranean Diet contains high-priced foods”) [51].

#### 3.2.6. Anthropometric and Sociodemographic Variables

Finally, anthropometric and sociodemographic information was self-reported by the respondents. Age, height, and weight were assessed as continuous variables, while others were assessed as categorical variables, including nationality, sex (i.e., “male”, “female”, and “other”), educational attainment (i.e., “lower secondary education or below”, “upper secondary education”, “Bachelor’s or equivalent level”, “postgraduate MSc or PhD”), income level (i.e., “A lot of difficulties getting to the end of the month”, “Some difficulty getting to the end of the month”, “No difficulty in reaching the end of the month”, “Manage to save money every month”, “I refuse to answer”), and geographical area of residence (6 categories: from “large urbanization area” to “remote rural area”). Using weight and height data, the subjects’ BMIs were computed and weight status was defined by applying the WHO’s standard cutoffs [52]. The subjects were asked to express the degree of responsibility for food purchasing and meal preparation (4 categories each: “always”, “often”, “sometimes”, and “never”), the habitual frequency of eating out and eating fast foods (7 categories each: from “Never” to “5 or more times per week”), the influence of religious beliefs and/or ethical concerns (4 categories: i.e., “Are your food habits affected”: “by religious beliefs”, “by ethical concerns”, “by religious beliefs and ethical concerns”, “not affected by any religious beliefs and/or ethical concerns”), and dietary regimes (i.e., omnivore, lacto-vegetarian, ovo-vegetarian, lacto–ovo vegetarian, pescatarian, vegan, flexitarian, and others).

### 3.3. Statistical Analysis

Descriptive and inferential statistics were performed on the surveyed variables. After having evaluated the internal consistency of the constructs with Cronbach’s alpha, we have computed the average values of the following variables, as already described in Section 3.2: picky eating, attitude toward the adoption of the Mediterranean Diet (MD), attitude toward the healthiness of foods, the food choice motives, and the drivers and barriers to the adoption of the MD. Adherence to the MD was assessed using the 14-item MEDAS questionnaire [50]. The respondents were then classified according to three categories of adherence to the MD calculated as follows: low MD adherence (scores from 0–5), medium MD adherence (scores from 6–9), high adherence (scores ≥ 10) [53]. The normality of the data distribution was evaluated and rejected through the Kolmogorov-Smirnov test. Thus, results were expressed as median and interquartile ranges (IQRs), whereas mean ± SD are reported in Appendix B. The non-parametric Kruskal–Wallis H test with the Bonferroni post hoc test was used to explore and compare differences between variables among subjects in different countries. ANOVA F-test was used to assess the equality of means across countries of the selected variables, whereas post-hoc Tukey’s Honest Significant Difference (HSD) test was used to assess the significance of differences between pairs of countries’ mean values (see Appendix B).

For predicting the antecedents of adherence to the MD and testing the hypotheses described in Figure 1, we have applied the Structural Equation Model (SEM) analysis on five models, one for each country. SEM allows to discern and assess the effects of a set of variables acting on a specified outcome via multiple causal pathways [54]. The effects of the variables were labelled as “beta” for the unstandardized coefficients, and β for the standardized coefficients. The goodness-of-fit of the models was tested by χ^2^ and degrees of freedom (*df*), Tucker-Lewis Index (TLI), comparative fix index (CFI), root mean square error of approximation (RMSEA), and standardized root mean square residual (SRMR), while the coefficient of determination (R^2^) was used to assess the explained variance of the endogenous variables. In particular, model adequacy and goodness is generally confirmed when CFI and TLI > 0.95, and SRMR and RMSEA < 0.08 [54]. Data analysis was conducted using the IBM^®^ SPSS^®^ Amos™ 24.0 software with the maximum likelihood estimator [55], and a *p* < 0.05 was considered as statistically significant.

## 4. Results

The description of results is organized, initially introducing the descriptive statistics, including the general food habits. Then, food choice motives followed by picky eating (presenting the correlation between its four domains) and adherence to the Mediterranean Diet (according to the MEDAS score) are presented. Finally, factors predicting adherence to the Mediterranean Diet and barriers and motives to adherence to the Mediterranean Diet are described.

### 4.1. Descriptive Statistics

The socio-demographics of the five samples are reported in the Table A1 in Appendix B. After excluding respondents who accessed the survey but did not complete it due to exclusion criteria or because they incorrectly selected the instructed response item (n = 4993), straight liners (n = 174), and fast respondents (n = 51), the total final sample consisted of 4025 subjects, which were almost equally distributed across the five countries (Greece n = 800, Italy n = 802, Morocco n = 803, Slovenia n = 806, and Tunisia n = 814). The total sample showed sex equity, made up of 49.6% females and 50.1% males. The mean age of the total sample was 42.6 ± 14.2 years old, while the lowest mean age was recorded in the Tunisian (37.1 ± 11.4 years old) and Moroccan (35.6 ± 10.9 years old) samples, which instead involved 44 (5.4% of the Tunisian sample) and 27 (3.4% of the Moroccan sample) individuals being 60 to 79 years old, respectively. The largest involvement of the 60–79 age category was recorded in Italy, where this category constituted 30% of the Italian sample. Concerning education, 40.8% of the total sample reported having an “upper secondary education” and only 6.1% indicated having a “Lower secondary education or below”. The “Small urban area” option was selected as the geographical area or origin by 23% of respondents, while, with respect to the level of income, “Some difficulty getting to the end of the month” option was indicated by 38.8% of the total sample.

The general food habits of respondents were assessed through general food-related questions. The level of involvement in food purchasing was registered as “high” for 52% of respondents in the total sample, while 39% of subjects indicated being “always involved” in meal preparation. Greece and Italy presented the highest level of involvement in both food purchasing and meal preparation. Conversely, Morocco and Tunisia registered the highest levels of occasional involvement (Figure 2a,b).

The influence of religious beliefs and/or ethical concerns over food habits registered the highest frequency at a country-level in Tunisia and Morocco, in which 55% of respondents reported to be affected by “religious beliefs”, 22% by “both religious and ethical”, and a lower % (2% for Morocco and 4% for Tunisia) by ethical concerns. Overall, 58% of the total sample was not affected by the aforementioned beliefs and concerns.

The “omnivore” option was the most frequently reported food diet, being selected by 89.3% of the total sample. Both in the total (72%) and single country samples, most of the respondents declared not to be under a low-calorie regime for weight loss or maintenance. For those that indicated being under a restricted calories diet regime, few of them reported being under the supervision of a healthcare professional, compared to those who ran the diet without this supervision.

Collectively, eating out of home was practiced “2–3 times per month” by 25.5% of respondents, followed by the option “less than once a month” selected by 25.4% of the total respondents. At a country level, Greek and Tunisian respondents reported the highest percentages of frequency of consumption of fast foods.

#### 4.1.1. Food Choice Motives and Picky Eating

The relative importance of food choice motives was evaluated considering the median values of a range of factors related to dietary choice for individuals (Table 1), whereas the mean values are reported in Table A2 (Appendix B). All scales showed as acceptable/good (α = 0.702; 0.844) up to good/excellent (α = 0.851; 0.909) internal consistency (Cronbach’s alpha). Further details are provided in Appendix B Table A3.

Overall, the highest median values were registered for the “natural content” and the “sensory appeal”, reflecting the same tendency of country levels where these factors accounted for almost all the highest values registered, compared to the other food choice motives. In Slovenia, the highest median value was registered for the “natural content” dimension, while the “price” factor was the second most reported dimension in Greece, according to the mean value 5.76 ± 1.07 (Table A2). In Morocco, Tunisia, and Greece, the “health” motive was also among the most important dimensions according to the median values. The lowest median values were registered for familiarity and weight control, compared to the other food choice motives, in both total sample and single country levels.

Regarding the sustainability-related dimensions, the general sustainability factors “ethical concerns” and “environmental welfare” registered the highest median values in all countries. Excluding the halal item, the “animal welfare” factor reported significantly higher median values in Italy. The same also applied for the general sustainability factor with significantly higher median scores reported in Italy and Morocco. The “local and seasonal” factor registered the lowest median values in all the countries compared to the other motives; in Italy and Slovenia, these items reported higher values compared to the other countries (Table 1).

Four domains of picky eating (PE) were investigated, i.e., meal presentation, food variety, meal disengagement, and taste aversion. In general, PE behaviors are relatively more endorsed in Morocco and Tunisia and less so in Slovenia and Italy. When considering the four dimensions, overall, taste aversion (e.g., rejection of bitter foods, sour foods, texture preference, etc.) is the facet of PE with the highest values across the five countries, followed by food variety (e.g., eating a limited number of food items, lack of food variety, etc.), and meal presentation (e.g., preference for specific food presentation, colors, specific sequence, etc.). Meal disengagement (e.g., mealtimes avoidance, etc.) is the least important factor in almost all countries (Table 1).

Overall, the outcomes of this section (FCQ, SUS-FCQ, and APEQ), investigating the main food choice motives and picky eating across countries, have shown the following (Table 1 and Table A2):Sensory appeal, health, and natural content were the most important food choice motives across the studied countries;Weight control and familiarity were the least important factors;Ethical concerns and environmental welfare were the most important sustainability food choice motives in all countries;Animal welfare was the least important sustainability food choice motive in Greece, Italy, and Slovenia and became relatively more important, especially in Italy, when the halal item was excluded;The local and seasonal dimension was the least relevant sustainability food choice motive in all countries;Picky eating was more endorsed in Morocco and Tunisia and relatively less so in Italy and Slovenia;Considering its four dimensions, taste aversion was the facet of PE with the highest values across the five countries, followed by food variety;Meal disengagement is the least important PE factor in almost all countries.

#### 4.1.2. Adherence to the Mediterranean Diet

Adherence to the MD was calculated according to the MEDAS screener responses on a 0–14 score range. In Morocco and Tunisia, the maximum score was 13 since the question of wine consumption was not included for cultural reasons (Figure 3).

Overall, results showed a medium-to-low adherence to the MD, ranging from the lowest median value registered in Slovenia, which was significantly lower compared to the other countries, as opposed to Italy and Morocco that showed the highest levels of adherence. Slovenia and Tunisia showed the highest percentages of individuals with poor adherence to the MD (39% and 19.4%, respectively), whereas consumers showing high MD adherence were more frequently found in Italy (16.5%) and Morocco (14.6%). Respondents in Greece and Morocco are more frequently in the category with medium adherence to the MD (74.6% and 73.6%, respectively) (Table 2).

### 4.2. Factors Predicting Adherence to the Mediterranean Diet

Separate structural equation models were performed to determine the effects of predictors to the adherence score to the MD across the five countries. The analysis was set based on the theoretical context, including as exogenous variables picky eating [42,43,44,48], attitude toward the adoption of the MD [41], interest in eating healthily [40], the food choice motives based on the Food Choice Questionnaire (FCQ) [30,31], and the sustainable food choice motives [38], as postulated in Figure 1. Table 3 provides the standardized coefficients (β), *p*-value, R-squared, and the model fit indices, whereas the unstandardized coefficients and standard errors are provided in Table A4 (see Appendix B).

As reported in Table 3, model fit indices showed an overall satisfactory goodness of fit for the models tested [54]. The models accounted for a share of the variance in adherence to the MD, ranging from 11% in Tunisia up to 26.6% in Slovenia, followed by Italy (22.1%), Greece (19.0%), and Morocco (12%). The results clearly indicated that attitude toward healthy eating was the most relevant and significant predictor of adherence to the MD in all countries. This means that consumers’ interest in eating healthily (e.g., “I always follow a healthy and balanced diet”, “It is important for me that my diet is low in fat”, etc.) is a relevant predictor of adherence to the MD. Furthermore, attitude toward the adoption of the MD was also a significant positive predictor of the MD adherence score in Slovenia, Italy, and Greece. In those countries, those more in favor of the adoption of the MD also have higher MD adherence scores. Conversely, this was not the case for Tunisia and Morocco. The coefficients associated with picky eating (Table 3) showed a negative effect of this exogenous variable over adherence to the MD in Slovenia, Italy, Morocco, and Tunisia. In Greece, this factor is negatively related with the respondents’ attitude toward the MD but not with the behavior itself. PE was also found to negatively influence the participants’ attitude toward the MD in Italy and Slovenia.

Among the food choice motives, health motives affected Moroccan, Slovenian, and (to a lesser extent) Tunisian respondents’ MD adherence. Natural content was associated with significant coefficients in Italy, where it was the strongest predictor; Tunisia and Morocco showed the positive effect of this motive on the MD adherence in these countries. Considering Greek and Slovenian respondents, both showed a positive effect of weight control over adherence to the MD. In Italy, the sensory appeal exhibited a negative effect on adherence to the MD. Price and convenience were negatively associated with adherence to the MD in Tunisia and Greece, respectively.

Looking at the sustainable food choice motives, the local and seasonality factors showed a positive effect on adherence to the MD in Greece and Morocco, whereas it was only slightly affecting the Italian respondents’ scores (*p* = 0.077). As indicated by the *p*-values reported in correspondence with general sustainability (Table 3) motives, no effects were shown on respondents’ MD adherence, whereas a positive effect was found on the attitude toward the MD in Italy.

### 4.3. Barriers and Motives to Adherence to the Mediterranean Diet

Among the barriers to the MD, health, restrictiveness, convenience, taste, food culture, affordability and access were investigated, while health, diet quality, applicability, lifestyle, affordability, and environment were evaluated as drivers considering the median values (Table 4), whilst the mean values are reported in Table A5 (Appendix B).

Overall, participants in all the Mediterranean countries involved perceived more benefits compared to barriers to the adoption of MD. Drivers’ average scores range from 4.13 (“Affordability” in Slovenia) to 6.08 (“Diet quality” in Greece), while barriers’ average scores ranged from 2.04 (“Taste” in Italy) to 4.03 (“Health” in Italy) (Table A5).

In general, diet quality (having healthier and more nutritious food, higher F&Vs, and lower meat consumption), applicability (tastier and more sustainable diet), and lifestyle (homemade, unprocessed, additive-free food, socialization, and family relationships) were identified as the most frequently reported advantages of the MD. Health benefits (positive effect on cholesterol, lowers LDL, and reduces health risks) were the second highest perceived driver in Slovenia. Considering the differences between countries, diet quality, applicability, lifestyle, and health benefits recorded the highest scores in Italy and Greece compared to the other countries (Table 4). The environmental (positive for environment, better carbon footprint, and local food) and affordability benefits (easier food access and lower-priced food) that were, respectively, the second least and the least perceived, received the highest scores in Italy. Overall, the perceived benefits of the MD had similar patterns within the Greek and Italian samples and within the Tunisian and Moroccan ones.

Among the factors limiting the adoption of the MD, health (contains allergenic foods) and affordability (contains high-priced foods) were the most relevant barriers to the MD for the consumers in all countries. In general, taste (contains unpleasant-tasting foods), food culture (conflict with cultural habits), and restrictiveness (insufficient food variety, restrictive, and difficult to diversity food recipes) were the least relevant barriers for the respondents in all countries. The differences between countries (Table 4) indicated that health was a highly perceived barrier in all countries except Slovenia and that affordability was strongly perceived as a barrier in all countries, with significantly lower values in Italy. Convenience (difficulty in terms of preparation and time-consumption) was relatively more strongly perceived as a barrier in Tunisia and Morocco, whereas access (limited options in shops and restaurants) was a relevant barrier in Slovenia and Morocco. In general, respondents in Italy and Greece reported the lowest perceived barriers to adopting the MD, while those in Morocco reported the highest.

## 5. Discussion

The results of our study showed that attitude toward healthy eating is a significant predictor of adherence to the MD in all the countries involved. This result supports H11 (Figure 1), confirming that those with a more favorable attitude toward healthy eating have a higher adherence to the MD, as also documented by Roininen et al. [40].

Similarly, a positive effect was observed for the exogenous variable “attitude towards the adoption of the MD”, being a forecaster of adherence to the MD itself. The results registered in Slovenia, Italy, and Greece confirm H10. These findings are coherent with the existing literature regarding the attitude–behavior relationship [39]. Considering the other two countries (Morocco and Tunisia) where lower non-significant paths have been found, adherence to the MD was not affected by consumers’ evaluation of adherence to the MD itself. In these countries, the food culture barrier, intended as the conflict between following the MD and cultural habits, registered its highest values, despite being one of the less relevant barriers among the other countries. Therefore, it is likely that for Tunisian and Moroccan respondents with a positive attitude toward the MD, it is not a sufficient driver for adhering more closely to this dietary behavior. Consequently, these results pose evidence for the attitude–behavior gap in the evaluation of healthy eating and the adoption of relative behavior, in particular when the behavior is not perfectly aligned with individuals’ cultural values. This finding seems to support the recent theoretical discussions about the importance of individuals’ goals in forming the motivation to consider performing a particular behavior [56]. For these countries, the recovery or rediscovery of traditional recipes in line with the MD and the gastronomic culture of these populations could represent a strategy to bridge this attitude–behavior gap.

Adult picky eating showed a negative effect on adherence to the MD with significant values in Slovenia, Tunisia, Morocco, and Italy; these findings support H12, confirming that those less prone to eating unfamiliar and novel foods show negative association with adherence to the MD. Menghi et al. [42] have also documented a negative link between adult PE and adherence to the MD in Italy. In our study, this negative influence was also found in the other countries, except for Greece. In this latter country, PE was a significant antecedent of consumers’ attitude toward the adoption of the MD. Therefore, in Greece, we found that the effect of adult PE on adherence to the MD is likely to be mediated by the consumers’ attitude. In other words, those who are unwilling to eat familiar foods or try novel foods have a more negative attitude toward the MD and, given this attitude, are also showing lower adherence to the MD.

Regarding the other exogenous variables included in the model, among food choice motives [30,31], the health dimension was found to be positively correlated with adherence to the MD in Morocco and Slovenia, confirming H1 for those countries. Researchers in the literature have indeed indicated a positive relation between the health dimension and both the consumption of vegetables and the association of the MD with health benefits [6,32]. Therefore, those giving more importance to those healthy aspects in food choices have a higher adherence to the MD.

Subjects paying more attention to the natural content showed higher adherence to the MD in Italy, Morocco and Tunisia, thus confirming H2 for those countries and the results of studies conducted in other countries [33]. H3 was confirmed in Greece and Slovenia, indicating that, for those two countries, a positive effect of the weight control dimension on the respondents’ adherence to the MD is observable, similarly to what was found in the US [33].

The familiarity factor is not a significant predictor of MD adoption; therefore, H7 was not supported by the results. Instead, the food choice motives related to the sensory appeal, price, and convenience have evidenced negative effects on adherence to the MD in different countries. The sensory appeal dimension negatively affected adherence to the MD in Italy, confirming H4. This means that those indicating a preference for food that smells nice, with a pleasant texture and that tastes well, exhibited a lower MD adherence. A weaker negative influence of this dimension was found in Morocco and Slovenia too. This result indicates that, in those countries, increasing the sensory appeal of the foods related with the MD could be associated with an increase in the MD score [34].

In Tunisia, consumers expressing a higher importance for the price factor, such as preference for cheaper food and good value for money, showed a lower adherence to the MD, thus supporting H6. Affordability was also found to be a significant barrier to the MD in all countries. As already found in France [35] and Italy [36], convenience, familiarity, and price have been identified as relevant barriers for the adoption of sustainable dietary behaviors in different Mediterranean countries. In Italy, another study has shown that the monthly expenditure of the MD is slightly higher in the overall budget compared to the current expenditure allocated to food by the Italian population, with a substantial difference in the distribution of budget according to the different food groups [17]. A higher need for food expenditure to achieve higher adherence to the MD was also found in Spain [57]; these results may suggest that, in the poorest regions or sub-population groups, where access to MD is less affordable in relative terms, fiscal measures should be proposed in the form of income supplements for the lowest income levels.

Similarly, Greek consumers that showed a higher preference for convenience dimension (e.g., easy-to-cook) were linked to a lower MD adoption, as postulated in H5. Convenience was also found to be a significant barrier to the MD in all countries. In Greece, the production of more convenient and easy-to-handle food (e.g., snacks) with ingredients linked to the MD could possibly increase the adherence to this dietary pattern. In Slovenia, the convenience factor positively affected the attitude toward the MD, indicating that ready-to-eat or easy-to-cook products with ingredients typical from the Mediterranean basin could increase the consumers’ attitude toward the MD and, in turn, lead to higher MD scores.

Looking at the sustainable food choice motives, the local and seasonal dimension was found to positively affect the MD adoption in Morocco and Greece. A weaker positive correlation was found in Italy too, thus confirming H9. Moroccan, Greek, and Italian consumers who expressed a higher attention to local, regional, and seasonal products showed a higher adherence score to the MD. This result confirms recent findings showing the relationship between adherence to the MD and sustainable dietary behaviors [37], as well as the revision and restructuring of the Mediterranean Diet Pyramid incorporating the sustainability and environmental impact of this dietary pattern [58]. However, H8 was not supported, since the positive effect of the general sustainability was only found in Italy on the consumers’ attitude toward the MD, not on the behavior itself. In other words, Italian consumers that are more interested in animal/environmental welfare and ethical concerns also have a more positive attitude toward the adoption of the MD; as a result, more favorable evaluation also exhibited higher MD scores.

Overall, our results demonstrate regional differences in the factors affecting, as well as in the barriers preventing, the adoption of the MD. In general, since attitudes toward healthy eating were found to be the most relevant predictor of adherence to the MD in the five countries, designing intervention strategies aimed at providing information to increase consumers’ awareness about the health impact of balanced and healthier dietary behavior is highly recommended. Although Sogari et al. [59] found that health messages did not influence US young adults to eat more whole grain food products, such informative strategies could be more effective in other population groups. A meta-analysis found that, when it comes to food products in particular, the use of gain frames (i.e., messages framed to promote the advantages of consuming a particular food) elicits stronger responses from consumers than the use of loss frames (i.e., messages framed to stress the negative outcomes of not consuming that particular food) [60]. Thus, promoting consumers’ information regarding the benefits of the MD adoption and stimulating their beliefs regarding the topic could contribute to increasing their attitude toward adherence to the MD and, in this way, the behavior itself. This might not be the case in countries such as Morocco and Tunisia, where this dimension was not found to affect the adoption to the MD. Given the variety of outcomes gained in the different countries involved regarding food choice motives, a tailored approach may be recommended to specifically promote some categories of less frequently consumed foodstuffs. Finally, given that the “variety of food consumption” is a pillar in the MD and having found a negative influence of PE on its adoption, strategies based on exposure could be a valuable option to promote adherence to the MD.

Other more practical approaches might be targeted to the implementation of a nutritional intervention program that introduces MD-inspired meals in workplace and school canteens. Studies have shown that such programs can successfully and sustainably introduce new dietary habits among working populations and adolescents with a minimal level of intervention [61,62]. Peer support interventions can also be applied as practical measures to promote the MD, in particular, for populations whose behavior is strongly influenced by subjective norm [59]. A 12-month randomized controlled trial targeting non-Mediterranean populations at high cardiovascular disease risk and with low MD adherence compared three intervention strategies: peer support, dietician-led, and minimal support groups. The study observed significant improvements in BMI, HbA1c levels, and blood pressure. These findings indicate that peer support strategies can facilitate dietary behavior changes toward adopting an MD [63].

To accommodate diverse dietary preferences, such as those of picky eaters, various meal delivery kits now include MD-based options specifically designed to adhere to the MD while also prioritizing convenience. Additionally, modern recipe books offering time-saving MD recipes can greatly benefit consumers looking for easy and quick meal options. Resources such as dietary guides from the Harvard School of Public Health provide examples of diet plans that follow MD principles, offering practical advice for incorporating the diet into daily routines [64].

Policy interventions have also proven crucial for promoting the MD, particularly among the lowest socioeconomic groups [57,65]. It has been suggested that the combination of policies, including taxes on specific foods aimed at discouraging unhealthy eating (e.g., sweetened beverages) and eliminating/reducing taxes (e.g., value-added tax) or introducing subsidies for healthy options (e.g., fruits and vegetables), can effectively promote healthier diets while providing support to the most vulnerable groups [66].

### Strengths, Limitations, and Future Directions

To the authors’ knowledge, no previous studies have assessed adherence to the MD in large representative samples across different European and African Mediterranean countries. Food choice motives and attitudinal variables associated with eating habits were also evaluated to provide a better overview of the factors facilitating or discouraging Mediterranean dietary behaviors in the studied countries. The results could possibly help in designing appropriate local and national policies and interventions to promote MD.

Despite the novelty of our research, some inherent limitations need to be outlined. A major limitation of this study is the lack of use of validated psychometric measures across countries, in particular for the APEQ and the SUS-FCQ. However, a partial validity of the used measures is supported by the values of Cronbach’s alphas. Nevertheless, future studies should address this issue. A second limitation has to do with the sampling method. While the sample size is adequate across countries, it is not fully representative of the studied population. For instance, the average education level is relatively high, with 39.2% of Tunisians holding a postgraduate MSc or PhD. Similarly, participants from Morocco and Tunisia are younger than those from other countries. It is also known that age affects adherence to the MD, albeit controversially. Moreover, we cannot fully exclude selection bias, in the sense that those who participated in the survey could be more interested in the topic discussed than the general population, thus giving a partial picture of the problem. Therefore, the results might have been influenced by this factor too and possible future investigations should engage older age groups in those countries to improve sample representativeness. Another limitation is related to the cross-sectional design, which does not allow for determining the temporal direction of the associations. Although this is a quite common practice [65], future studies should apply longitudinal studies to predict the prospective behavior. Finally, given the relatively low variance explained by the models across all countries, novel studies are needed to identify a more comprehensive regional list of predictors including, e.g., variables widely applied such as subjective norms, perceived behavioral control, and other psycho-social variables.

Future research direction should consider factors such as the demographics, attitudes, nutrition literacy, and psychosocial traits of consumers that significantly influence the likelihood of adopting the MD principles [28]. However, the evidence remains inconclusive for certain factors. For instance, the relationship between age, gender, and MD adherence is still debated. While some studies suggest that being female and older is associated with higher adherence to the MD, others report contradictory findings or no gender differences [14]. Nevertheless, further insights on how lifestyle behaviors and age-related factors influence MD adherence could enhance the discussion [67]. Likewise, although the positive impact of awareness and literacy regarding the MD’s health benefits on dietary variety and quality is well documented [28], less is known about the psychological traits that may impede adherence. A significant body of the literature suggests that traits associated with selective eating negatively affect habitual intake of prototypical MD food groups [68]. However, previous studies reporting such negative effects are either limited to childhood [69] or to a specific geographical area [42]. Thus, it remains unclear as to whether selective eating traits contribute equally to the observed decline in MD adherence among adults across the Mediterranean region.

## 6. Conclusions

Given the increasing importance of achieving health and sustainable goals, the promotion of more sustainable dietary behavior, including the MD, assumes a central role. By preventing the abandonment of such a diet and implementing its adoption, especially in the Mediterranean countries, it is therefore crucial and is strictly linked to the understanding of factors predicting such behavior.

The present study confirmed, from a cross-country perspective, the relative importance of specific food choice motives (positively or negatively affecting the adoption of the MD) and of other attitudinal factors (i.e., attitudes toward the healthfulness of food, attitudes toward the MD, and adult picky eating) in attaining this goal.

Overall, the research reported a medium-to-low adherence score to the MD, with a largely prevalent share of low to medium MD adherence values for consumers registered in the five countries (Greece, Italy, Morocco, Slovenia, and Tunisia). Attitudes toward healthy eating were found to be the most relevant predictor of adherence to the MD in the five countries. This result suggests that policies or intervention strategies aiming at increasing consumers’ awareness about the health impact of balanced dietary behavior would likely increase adherence to the MD. Attitude toward the adoption of the MD is also a significant predictor in the European countries involved in the study but not in the Northern African ones, whereas pick eating negatively affects adherence to the MD in all countries except Greece. Finally, our results demonstrate regional differences in the factors affecting, as well as in the barriers preventing, the adoption of the MD. Given the variety of outcomes across the five countries regarding the influence of food choice motives on adherence to the MD, a tailored approach may be recommended for policy design.

## Figures and Tables

**Figure 1 nutrients-16-02405-f001:**
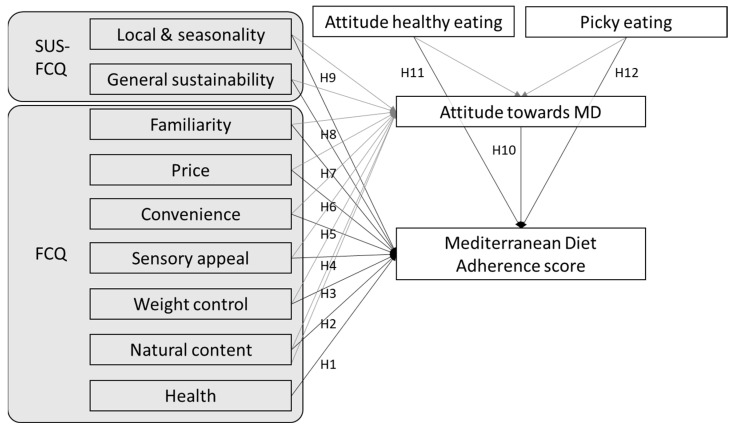
Hypothetical model linking food choice motives with attitude and behavior toward adherence to the Mediterranean Diet.

**Figure 2 nutrients-16-02405-f002:**
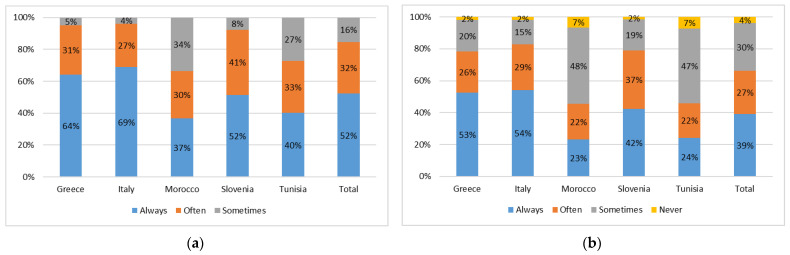
Level of involvement (%) in (**a**) food purchasing and (**b**) meal preparation within each country (Greece, Italy, Morocco, Slovenia, and Tunisia) and the total sample (total).

**Figure 3 nutrients-16-02405-f003:**
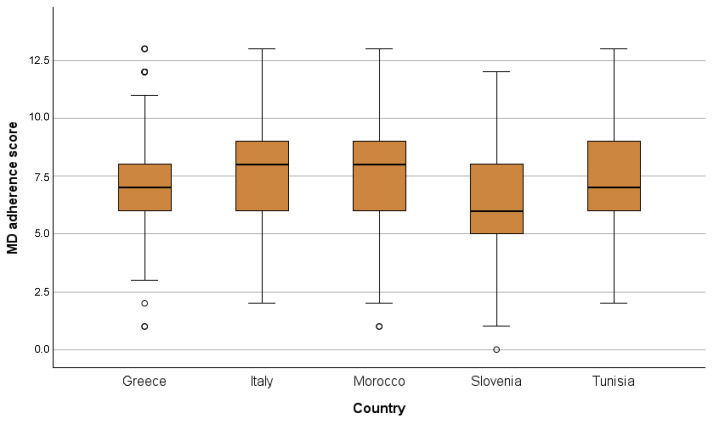
Adherence the Mediterranean Diet (MD) scores; box-plots (MD adherence scores ranging from 0 to 14, except for Morocco and Tunisia from 0 to 13).

**Table 1 nutrients-16-02405-t001:** Food Choice Questionnaire (FCQ) and Sustainable Food Choice Questionnaire (SUS-FCQ) motives, Adult Picky Eating Questionnaire (APEQ), Attitude toward Mediterranean Diet (AMD), and Health and Taste Attitude Scales (HTAS), median values (IQR) across the five countries.

	Greece (*n* = 800)	Italy (*n* = 802)	Morocco (*n* = 803)	Slovenia (*n* = 806)	Tunisia (*n* = 814)	Total (*n* = 4025)
**FCQ**						
Health	6.0 (5.0–6.9) ^a^	5.7 (5.0–6.7) ^a^	6.3 (5.3–7.0) ^b^	5.7 (4.7–6.7) ^a^	6.0 (5.0–7.0) ^a^	6.0 (5.0–7.0)
Convenience	5.3 (4.3–6.0) ^a^	5.3 (4.7–6.0) ^a^	5.3 (4.3–6.3) ^a^	5.3 (4.3–6.0) ^a^	5.0 (5.0–7.0) ^b^	5.3 (4.3–6.0)
Sensory appeal	6.0 (5.3–6.7) ^a,b^	6.0 (5.3–6.7) ^a^	6.3 (5.3–7.0) ^b^	5.7 (5.0–6.3) ^c^	6.0 (5.0–7.0) ^a,b^	6.0 (5.0–6.7)
Natural content	6.0 (5.0–7.0) ^a,b^	6.0 (5.0–7.0) ^a^	6.0 (5.0–7.0) ^a^	6.0 (4.7–6.7) ^b^	6.0 (4.7–7.0) ^a,b^	6.0 (5.0–7.0)
Price	6.0 (5.0–6.7) ^a^	5.7 (4.7–6.3) ^b^	5.7 (4.7–6.3) ^b^	5.7 (4.7–6.3) ^b^	5.3 (4.7–6.3) ^b^	5.7 (4.7–6.3)
Weight control	5.3 (4.3–6.3) ^a^	5.0 (4.0–6.0) ^b^	5.0 (4.0–6.0) ^b^	4.7 (3.7–5.7) ^c^	4.7 (3.3–5.7) ^c^	5.0 (4.0–6.0)
Familiarity	5.0 (4.3–6.0) ^a^	4.7 (3.7–5.3) ^b^	5.0 (4.0–6.0) ^c^	5.0 (4.3–6.0) ^a^	4.7 (3.3–5.3) ^b^	5.0 (4.0–5.7)
**SUS-FCQ**						
General sustainability (+halal) ^1^	5.4 (4.6–6.1) ^a^	5.7 (4.9–6.3) ^b^	6.1 (5.1–6.9) ^c^	5.2 (4.1–6.1) ^a^	6.0 (5.1–6.7) ^c^	5.7 (4.7–6.4)
General sustainability ^2^	5.7 (4.8–6.7) ^a^	6.0 (5.2–6.8) ^b^	6.0 (5.0–6.8) ^b,c^	5.7 (4.5–6.5) ^a^	6.0 (4.8–6.7) ^a,c^	5.8 (4.8–6.7)
Animal welfare (+halal) ^1^	5.0 (4.0–5.7) ^a^	5.0 (4.0–6.0) ^a^	6.0 (5.0–7.0) ^b^	4.7 (3.3–6.0) ^a^	6.0 (5.0–7.0) ^b^	5.3 (4.0–6.3)
Animal welfare ^2^	5.5 (4.0–7.0) ^a^	6.0 (5.0–7.0) ^b^	5.5 (4.0–7.0) ^a^	5.5 (4.0–7.0) ^a^	5.5 (4.0–7.0) ^a^	5.5 (4.0–7.0)
Ethical concern	6.0 (5.0–7.0) ^a^	6.0 (5.0–7.0) ^b^	6.5 (5.0–7.0) ^b^	6.0 (4.5–7.0) ^a^	6.0 (5.0–7.0) ^b^	6.0 (5.0–7.0)
Environmental welfare	6.0 (5.0–7.0) ^a^	6.0 (5.0–7.0) ^b^	6.0 (5.0–7.0) ^b^	6.0 (4.5–7.0) ^a^	6.0 (5.0–7.0) ^b^	6.0 (5.0–7.0)
Local and seasonal	5.3 (4.3–6.0) ^a^	5.7 (4.7–6.3) ^b^	5.0 (4.0–6.0) ^a^	5.3 (4.3–6.3) ^b^	4.7 (3.3–5.7) ^c^	5.3 (4.0–6.0)
**APEQ**						
Picky eating	2.3 (1.9–2.7) ^a^	2.2 (1.7–2.6) ^a,b^	2.8 (2.4–3.1) ^c^	2.1 (1.8–2.5) ^b^	2.7 (2.4–3.0) ^c^	2.4 (2.0–2.8)
Meal presentation	2.3 (1.9–2.7) ^a^	2.1 (1.6–2.6) ^b^	2.7 (2.3–3.1) ^c^	2.0 (1.6–2.4) ^d^	2.7 (2.3–3.1) ^c^	2.4 (1.9–2.9)
Food variety	2.3 (1.8–2.8) ^a^	2.3 (1.5–2.8) ^a^	2.8 (2.3–3.0) ^b^	2.3 (1.8–2.8) ^a^	2.8 (2.3–3.0) ^b^	2.5 (1.8–3.0)
Meal disengagement	2.0 (1.3–2.7) ^a^	1.7 (1.0–2.3) ^b^	2.3 (2.0–3.0) ^c^	2.0 (1.7–2.7) ^a^	2.3 (2.0–3.0) ^c^	2.0 (1.7–2.7)
Taste aversion	2.3 (1.8–2.8) ^a^	2.4 (1.8–3.0) ^a^	3.0 (2.5–3.3) ^b^	2.2 (1.8–2.7) ^c^	3.0 (2.5–3.3) ^b^	2.5 (2.0–3.0)
**AMD**						
Attitude toward MD	7.0 (6.0–7.0) ^a^	6.8 (6.0–7.0) ^a^	5.5 (4.3–6.8) ^b^	6.0 (5.0–7.0) ^c^	6.0 (4.3–6.8) ^b^	6.3 (5.0–7.0)
**HTAS**						
Attitude healthy eating	4.6 (4.0–5.3) ^a^	4.8 (4.1–5.5) ^b^	4.8 (4.3–5.5) ^b^	4.4 (3.9–5.1) ^c^	5.0 (4.3–5.6) ^d^	4.8 (4.1–5.4)

The food choice motives scale and the sustainable food choice motives scale were recorded on a 7-point scale (1—Not at all important; 7—Extremely important). The picky eating items were recorded on a 5-point scale (1—Never; 2—Rarely; 3—Sometimes; 4—Often; 5—Always). Attitude toward the adoption of the MD and attitude toward healthy eating were recorded on 7-point scales. Note: ^1^ includes the item “Is produced in Halal way”; ^2^ does not include the item “Is produced in Halal way”. Medians followed by a common letter are not significantly different, while different letters indicate statistically significant differences between countries (Kruskal–Wallis H test with the Bonferroni post hoc test, *p* < 0.05).

**Table 2 nutrients-16-02405-t002:** Adherence to the Mediterranean Diet (MD) scores across the five countries: mean, standard deviation (SD), median, and interquartile range (IQR). Levels of adherence to the MD [53].

		Greece (*n* = 800)	Italy (*n* = 802)	Morocco (*n* = 803)	Slovenia (*n* = 806)	Tunisia (*n* = 814)	Total (*n* = 4025)
Mean (SD)		7.17 (1.79) ^b^	7.67 (1.89) ^a^	7.62 (1.84) ^a^	6.06 (2.05) ^c^	7.21 (1.91) ^b^	7.14 (1.98)
Median (IQR)		7.00 (6.00–8.00) ^b^	8.00 (6.00–9.00) ^a^	8.00 (6.00–9.00) ^a^	6.00 (5.00–8.00) ^c^	7.00 (6.00–9.00) ^b^	7.00 (6.00–8.00)
Low adherence (scores 0–5)	N	132	95	95	314	158	794
	%	16.5	11.8	11.8	39.0	19.4	19.7
Medium adherence (scores 6–9)	N	597	575	591	458	564	2785
	%	74.6	71.7	73.6	56.8	69.3	69.2
High adherence (scores ≥ 10)	N	71	132	117	34	92	446
	%	8.9	16.5	14.6	4.2	11.3	11.1

Note: Means followed by a common letter are not significantly different; different letters indicate statistically significant differences between countries (post-hoc Tukey HSD test, *p* < 0.001). Medians followed by a common letter are not significantly different, while different letters indicate statistically significant differences between countries (Kruskal–Wallis non parametric test, *p* < 0.001).

**Table 3 nutrients-16-02405-t003:** Structural equation models predicting the attitude toward the Mediterranean Diet (MD) via an adherence score to the MD: standardized coefficients (β), *p*-value, R-squared, and model fit indices across the five countries.

Predictors	Greece (*n* = 800)		Italy (*n* = 802)		Morocco (*n* = 803)		Slovenia (*n* = 806)		Tunisia (*n* = 814)	
**Attitude toward MD**	**β**	** *p* **	**β**	** *p* **	**β**	** *p* **	**β**	** *p* **	**β**	** *p* **
Health	0.001	0.978	−0.006	0.899	0.047	0.309	0.024	0.591	0.109	0.008
Natural content	0.139	0.004	−0.033	0.503	0.009	0.847	0.114	0.019	0.083	0.049
Weight control	0.041	0.370	−0.005	0.904	0.000	0.997	−0.078	0.088	−0.050	0.242
Sensory appeal	0.076	0.055	0.056	0.158	−0.019	0.665	0.076	0.061	−0.045	0.254
Convenience	−0.065	0.108	−0.021	0.604	0.077	0.067	0.104	0.013	0.017	0.674
Price	0.027	0.486	−0.046	0.236	0.019	0.646	−0.071	0.073	0.066	0.088
Familiarity	0.006	0.885	0.143	<0.001	−0.001	0.980	−0.033	0.405	0.013	0.748
General sustainability	−0.006	0.883	0.164	<0.001	0.000	0.992	0.003	0.948	−0.055	0.182
Local and seasonal	0.086	0.047	0.073	0.097	0.026	0.553	0.091	0.047	0.023	0.584
Attitude healthy eating	0.038	0.400	0.058	0.219	0.021	0.611	0.092	0.054	0.046	0.294
Picky eating	−0.090	0.010	−0.115	0.002	0.022	0.542	−0.090	0.012	−0.046	0.207
R^2^	0.095		0.096		0.016		0.083		0.035	
**Adherence to the MD**	**β**	** *p* **	**β**	** *p* **	**β**	** *p* **	**β**	** *p* **	**β**	** *p* **
Health	0.049	0.278	0.052	0.213	0.111	0.010	0.112	0.005	0.072	0.071
Natural content	0.079	0.081	0.220	<0.001	0.087	0.043	0.061	0.160	0.130	0.001
Weight control	0.109	0.012	0.051	0.223	0.063	0.115	0.092	0.024	0.045	0.275
Sensory appeal	−0.033	0.371	−0.117	0.002	−0.077	0.057	−0.066	0.068	−0.028	0.463
Convenience	−0.099	0.009	−0.066	0.080	−0.032	0.426	−0.047	0.209	−0.029	0.453
Price	−0.030	0.402	−0.050	0.165	−0.067	0.090	−0.001	0.979	−0.110	0.003
Familiarity	−0.057	0.144	−0.047	0.222	−0.032	0.425	−0.033	0.360	0.022	0.589
General sustainability	−0.016	0.677	−0.067	0.113	0.005	0.906	0.030	0.469	−0.019	0.634
Local and seasonal	0.082	0.046	0.073	0.077	0.102	0.013	0.046	0.262	0.057	0.159
Attitude toward MD	0.073	0.029	0.114	<0.001	0.032	0.333	0.151	<0.001	0.050	0.139
Attitude healthy eating	0.238	<0.001	0.218	<0.001	0.178	<0.001	0.260	<0.001	0.148	<0.001
Picky eating	−0.030	0.371	−0.070	0.044	−0.072	0.039	−0.094	0.003	−0.079	0.024
R^2^	0.190		0.221		0.120		0.266		0.110	
Model fit indices:										
χ^2^ (df)	9.061 (9)		19.801 (7)		35.224 (8)		15.428 (10)		12.362 (5)	
CFI	1.000		0.996		0.988		0.998		0.996	
TLI	1.000		0.953		0.886		0.986		0.942	
RMSEA (90% CI)	0.003 (0.000–0.040)		0.048 (0.024–0.073)		0.065 (0.044–0.088)		0.026 (0.000–0.050)		0.043 (0.012–0.073)	
SMRM	0.018		0.031		0.039		0.020		0.015	

**Table 4 nutrients-16-02405-t004:** Main drivers and barriers to the Mediterranean Diet (MD): Median values (IQR).

Drivers and Barriers to the MD	Greece (*n* = 800)	Italy (*n* = 802)	Morocco (*n* = 803)	Slovenia (*n* = 806)	Tunisia (*n* = 814)	Total (*n* = 4025)
**Drivers**						
Health (positive effect on cholesterol, lowers LDL, and reduces health risks)	6.0 (5.0–6.7) ^a^	6.0 (5.3–6.3) ^a^	5.0 (4.3–6.0) ^b^	5.7 (4.7–6.0) ^c^	5.0 (4.0–6.0) ^d^	5.7 (4.7–6.3)
Diet quality (healthier, more nutritious food, higher F&Vs, and lower meat consumption)	6.0 (5.7–7.0) ^a^	6.0 (5.7–6.7) ^a^	5.7 (5.0–6.3) ^b^	6.0 (5.0–6.3) ^b^	5.7 (4.7–6.0) ^c^	6.0 (5.0–6.7)
Applicability (tastier and more sustainable than other diets)	6.0 (5.0–7.0) ^a^	6.0 (6.0–7.0) ^a^	6.0 (5.0–6.0) ^b^	5.0 (4.0–6.0) ^c^	6.0 (5.0–7.0) ^b^	6.0 (5.0–7.0)
Lifestyle (homemade, unprocessed, additive-free, socialization, and family relations)	6.0 (5.3–6.3) ^a^	6.0 (5.0–6.3) ^a,d^	5.7 (5.0–6.3) ^b^	5.3 (4.3–6.0) ^c^	5.7 (5.0–6.3) ^b,d^	5.7 (5.0–6.3)
Affordability (easier food access, and lower-priced food)	5.0 (4.5–6.0) ^a^	5.5 (4.5–6.0) ^b^	5.0 (4.5–6.0) ^b^	4.0 (3.5–5.0) ^c^	5.0 (4.5–6.0) ^a,b^	5.0 (4.0–6.0)
Environment (positive for environment, better carbon footprint, and local food)	5.3 (4.5–6.0) ^a^	5.5 (4.8–6.3) ^b^	5.3 (4.8–6.0) ^a^	4.8 (4.0–5.5) ^c^	5.3 (4.5–6.0) ^a^	5.3 (4.5–6.0)
**Barriers**						
Health (contain allergenic foods)	4.0 (3.0–5.0) ^a^	4.0 (3.0–5.0) ^a^	4.0 (3.0–4.5) ^a^	4.0 (2.5–4.5) ^b^	4.0 (3.0–5.0) ^a^	4.0 (3.0–4.5)
Restrictiveness (insufficient food variety, restrictive, and difficult diversify food recipes)	2.3 (1.7–3.3) ^a^	2.0 (1.3–3.0) ^b^	3.3 (2.3–4.3) ^c^	3.0 (2.0–4.0) ^d^	3.0 (2.0–4.0) ^d^	2.7 (2.0–3.7)
Convenience (difficult to prepare and time-consuming)	3.0 (2.0–4.0) ^a^	3.0 (2.0–4.0) ^a^	4.0 (3.0–5.0) ^b^	4.0 (2.0–4.0) ^c^	4.0 (2.0–5.0) ^b,c^	3.0 (2.0–5.0)
Taste (contains unpleasant-tasting foods)	2.0 (1.0–3.0) ^a^	2.0 (1.0–2.0) ^b^	3.0 (2.0–4.0) ^c^	2.0 (2.0–4.0) ^d^	2.0 (1.0–3.0) ^e^	2.0 (1.0–3.0)
Food culture (conflict with cultural habits)	2.0 (2.0–4.0) ^a^	2.0 (1.0–3.0) ^b^	3.0 (2.0–5.0) ^c^	2.0 (1.0–4.0) ^a^	2.0 (2.0–4.0) ^a^	2.0 (1.0–4.0)
Affordability (contains high-priced foods)	4.0 (3.0–5.0) ^a^	3.0 (2.0–4.0) ^b^	4.0 (3.0–5.0) ^a^	4.0 (3.0–5.0) ^a^	4.0 (3.0–5.0) ^a^	4.0 (2.0–5.0)
Access (limited options in shops and restaurants)	3.0 (2.0–4.0) ^a^	2.0 (1.0–3.5) ^b^	4.0 (3.0–5.0) ^c^	4.0 (3.0–5.0) ^c^	3.5 (2.0–4.5) ^d^	3.5 (2.0–4.5)

Notes: The items were all recorded on a 7-point scale (1—Not at all important; 7—Extremely important). Medians followed by a common letter are not significantly different, while different letters indicate statistically significant differences between countries (Kruskal–Wallis H test with the Bonferroni post hoc test, *p* < 0.05).

## Data Availability

The data presented in this study are available on request from the corresponding author.

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
