# Peer review of "Drivers and Barriers Influencing Adherence to the Mediterranean Diet: A Comparative Study across Five Countries"

_nutrients, 2024, doi:10.3390/nu16152405_

Round 1

Reviewer 1 Report

Comments and Suggestions for Authors

Dear Editor,

I carefully read the manuscript "Drivers and barriers influencing the adherence to the Mediterranean Diet: A comparative study across five countries". My comments and suggestions for the authors are the following:

 - The authors stated that data collection was conducted in all the five countries by an external agency. Then, who collected the informed consent of the participants?

 - Line 174: Are the authors referring to gender or sex here?

 - Line 177: "applied the definition of adults based on Pubmed". I guess there are better references than pubmed in this case... or not?

 - Line 180: The authors should include more information as regard the "quality check".

 - Line 182: "Subjects presenting straight-line patterns followed by fast responders were excluded from the final sample, as they were deemed to be careless". I believe that this judgment is absolutely arbitrary and that all collected questionnaires should be considered. excluding questionnaires from the analysis increases the risk of bias.

 - Line 191: "An online questionnaire was designed on a theory-driven approach". How did the authors collect informed consent if the questionnaire was administered online?

 - Line 293: The authors should specify which normality test they performed.

 - The limitations of the study should be further and more deeply discussed by the authors.

Author Response

Dear Editor,

I carefully read the manuscript "Drivers and barriers influencing the adherence to the Mediterranean Diet: A comparative study across five countries". My comments and suggestions for the authors are the following:

Authors – the authors want to thank Reviewer#1 for his/her valuable suggestions, which allowed to improve the manuscript in terms of clarity and quality.

R#1 – 1. The authors stated that data collection was conducted in all the five countries by an external agency. Then, who collected the informed consent of the participants?

Authors – Thanks for this comment. To be part of the study, respondents were asked to accept the informed consent, which was shown at the beginning of the survey. The survey could be answered only by those subjects who agreed with the described conditions. The external agency collected online both data and informed consent. The text has been integrated accordingly (see Section 3.1, page 5, lines 186-190).

R#1 – 2. Line 174: Are the authors referring to gender or sex here?

Authors – Many thanks for addressing this point. Actually, the authors referred to sex. The text has been amended accordingly.  

R#1 – 3. Line 177: "applied the definition of adults based on Pubmed". I guess there are better references than pubmed in this case... or not?

Authors – Thanks for this comment. We amended the text specifying that we included adults and aged people (66-79 y), according to the classification provided by PubMed (see Section 3.1, page 5, lines 192-193). PubMed has been used as a reference to include older adults as they are a relevant share of the population in the involved European countries (60-79 years old subjects are 29-30% of the population living in Italy, Greece and Slovenia). 

R#1 – 4. Line 180: The authors should include more information as regard the "quality check".

Authors – Many thanks for this comment. Actually, the procedure was not clear enough so far. The text has been integrated for clarity (see Section 3.1, page 5, lines 198-200 and 205-210). As a priori screening, an instructed response item was inserted in a long matrix (i.e. to demonstrate that you are not a robot, for this statement, please select “Strongly agree”). As a posteriori screening, straight liners were identified based on the answers given to two question matrices (A, B) that included two items with opposite meanings:

  • A1) The healthiness of food has little impact on my food choices.
  • A2) I am very particular about the healthiness of food I eat.

  • B1) Mediterranean Diet contains lower-priced foods.
  • B2) Mediterranean Diet contains high-priced foods.

Indeed, for consistency, careful respondents should select different answers within those matrices, resulting in a standard deviation different to 0. Therefore, those respondents for which the standard deviation was equal to 0 for such matrix questions were excluded. Secondly, after removing straight liners, the second criterion applied was the time spent to complete the survey. The median time spent by subject who responded to the whole survey in a maximum of 60 minutes was used as reference, and those who responded in less than 40% of this value were eliminated as considered respondents providing low quality data. The criterion used to identify fast respondents was similar to the strategy applied in other studies (e.g., https://doi.org/10.1016/j.foodqual.2023.104971).

R#1 – 5. Line 182: "Subjects presenting straight-line patterns followed by fast responders were excluded from the final sample, as they were deemed to be careless". I believe that this judgment is absolutely arbitrary and that all collected questionnaires should be considered. excluding questionnaires from the analysis increases the risk of bias.

Authors – Many thanks for this comment. The data quality check was carefully addressed. Recommended actions to exclude low quality data were followed to avoid arbitrary judgment as much as possible. Indeed, only subjects who provided inconsistent data in relation to the two question matrixes mentioned in the text and those who answered in a very short time were eliminated. Please refer to the authors’ answer to comment number 4 for detailed information about straight liners and fast respondents identification. Additional information has been added to the text for clarity (see Section 3.1, page 5, lines 205-210).

R#1 – 6. Line 191: "An online questionnaire was designed on a theory-driven approach". How did the authors collect informed consent if the questionnaire was administered online?

Authors – Please refer to the authors’ answer to comment number 1.

R#1 – 7. Line 293: The authors should specify which normality test they performed.

Authors – Thanks a lot for this comment. The normality of the data distribution was evaluated and then rejected for most of the variables through the Kolmogorov-Smirnov test. While the "normality assumption" should be considered irrelevant when it comes to evaluating inferences in large sample sizes (like our group; https://doi.org/10.1016/j.pmrj.2012.10.013), we still favored a more conservative approach and used non-parametric tools to measure the cross-country comparison. Thus, we decided to include in the main text the results expressed as median and interquartile ranges (IQRs) (see Table 1, 2 and 4), while reporting the original mean ± SD in the Appendix (see Tables A2 and Table A5). Then, the non-parametric Kruskal-Wallis H test with Bonferroni post hoc test was used to explore and compare the differences between variables among subjects in different countries. See new Tables 1, 2 and 4.

R#1 8. The limitations of the study should be further and more deeply discussed by the authors.

Authors – Thanks for this comment. We have integrated the discussion of the limitations. We have added a new section (i.e. the Section 5.1) including the strengths, limitations and future directions. In particular, we’ve emphasised the potential selection bias; indeed, we cannot exclude that those who responded the survey may be more interested in the topic discussed than the general population, thus giving a partial picture of the problem. This issue was addressed in the discussions (see Section 5.1 “Strengths, limitations and future directions”, lines 718-720).

Reviewer 2 Report

Comments and Suggestions for Authors

This manuscript investigates the drivers and barriers influencing adherence to the Mediterranean Diet (MD) across five countries: Greece, Italy, Morocco, Slovenia, and Tunisia. Using data from 4,025 individuals, the study employs Structural Equation Modeling to assess the impact of various factors on MD adherence. Key findings indicate medium-to-low adherence overall, with the highest in Italy and Morocco and the lowest in Slovenia. Positive attitudes towards food healthiness were the strongest predictors of adherence, while picky eating negatively influenced adherence. The study underscores the need for country-specific interventions to promote MD adherence. It is a very interesting article, my congratulations.

Here are some minor issues that should be addresses.

Introduction

The introduction provides a comprehensive background on the Mediterranean Diet, its health benefits, and the decline in adherence. However, it could benefit from a more explicit statement of the research gap that this study aims to fill. 

Please, clarify the novelty of the study by explicitly stating how it differs from previous research on the Mediterranean Diet.

Methods

The methodology section is detailed, describing the data collection process, sample characteristics, and measures used.

The section on data quality checks is comprehensive, but it would be helpful to include information on the response rate and how non-response bias was addressed. 

Provide more detail on the translation and back-translation process for the questionnaire to ensure linguistic validity.

Results

The results are presented clearly with appropriate use of tables and figures. The use of SEM to test the hypotheses is appropriate and well-explained.

However, the presentation of the results could be more concise. Some sections are repetitive and could be streamlined. Highlight the key findings in a summary table or bullet points at the end of the results section for better readability.

Discussion

The discussion interprets the findings in the context of existing literature and highlights the implications for public health and policy. The identification of country-specific factors is a valuable contribution.

However, it could benefit from a more critical examination of the study's limitations and potential sources of bias. 

Discuss the generalizability of the findings and suggest areas for future research. For instance, integrating insights on how lifestyle behaviors and age-related factors influence MD adherence could enhance the discussion (https://www.mdpi.com/2072-6643/15/23/4892, https://www.mdpi.com/2072-6643/15/23/4892). 

Conclusion

The conclusion succinctly summarizes the main findings and their implications. It effectively reinforces the need for tailored interventions to promote MD adherence.

Proposing specific policy recommendations based on the study's findings would be a plus.

Comments on the Quality of English Language

The manuscript is well-written, but minor edits for clarity and flow would be beneficial.

Author Response

This manuscript investigates the drivers and barriers influencing adherence to the Mediterranean Diet (MD) across five countries: Greece, Italy, Morocco, Slovenia, and Tunisia. Using data from 4,025 individuals, the study employs Structural Equation Modeling to assess the impact of various factors on MD adherence. Key findings indicate medium-to-low adherence overall, with the highest in Italy and Morocco and the lowest in Slovenia. Positive attitudes towards food healthiness were the strongest predictors of adherence, while picky eating negatively influenced adherence. The study underscores the need for country-specific interventions to promote MD adherence. It is a very interesting article, my congratulations.

Here are some minor issues that should be addressed.

Authors – The authors wish to thank Reviewer#2 for his/her positive evaluation, as well as for his/her suggestions which allowed us to improve the manuscript.

R#2 - Introduction

The introduction provides a comprehensive background on the Mediterranean Diet, its health benefits, and the decline in adherence. However, it could benefit from a more explicit statement of the research gap that this study aims to fill. Please, clarify the novelty of the study by explicitly stating how it differs from previous research on the Mediterranean Diet.

Authors – Thanks for this comment. The section was implemented with references to a recent systematic review (https://doi.org/10.3390/nu14204314) on drivers and barriers to adherence to the MD, including the research gap considering the available studies, as well as the review already cited (https://doi.org/10.1007/s00394-022-02885-0) that also indicated the need for investigating the main determinants of adherence to the MD in Mediterranean non-European countries. See revision at page 2, lines 83-96).

R#2 - Methods

The methodology section is detailed, describing the data collection process, sample characteristics, and measures used. The section on data quality checks is comprehensive, but it would be helpful to include information on the response rate and how non-response bias was addressed. Provide more detail on the translation and back-translation process for the questionnaire to ensure linguistic validity.

Authors – Many thanks for this question, which is fully appropriate. Unfortunately, it was not possible to calculate the response rate intended as the number of completed responses divided by the number of surveys sent as the latter is not known by the authors. Indeed, the invitation to complete the survey was externally managed by a marketing company. Nonetheless, additional information has been added to the text. This includes a) the number of respondents who accessed the survey but did not complete it due to exclusion criteria or because they wrongly selected the instructed response item applied as a priori screening (i.e., to demonstrate that you are not a robot, for this statement, please select “Strongly agree”); b) the number of respondents exhibiting straight lines responding and fast respondents (see Section 4.1, page 9, lines 360-363). The text has been integrated accordingly and a clarification about translation and back-translation process have been provided (see Section 3.2, page 6, lines 220-224).

Moreover, we fully agree with the referee that potentially a selection bias was present in our study. Indeed, those who participated in the survey might have been more interested in the topic discussed than the general population, thus giving a partial picture of the problem. This issue was addressed in the new Section “5.1 Strengths, limitations and future directions” (see lines 718-720).

Finally, we’ve included a sentence on the approval from the local institutional Ethical Committee (REB - Research Ethics Board, 28-2023-N, March 29th, 2023) and that data was analyzed anonymously (respectively, lines 181-183, and lines 188-190).

R#2 - Results

The results are presented clearly with appropriate use of tables and figures. The use of SEM to test the hypotheses is appropriate and well-explained. However, the presentation of the results could be more concise. Some sections are repetitive and could be streamlined. Highlight the key findings in a summary table or bullet points at the end of the results section for better readability.

Authors – Thanks for this comment. A bullet point summary has been added (see Section 4.1.1, lines 433-450) to highlight the major outcomes of section 4.1.1 and facilitate the interpretation of Table 1.

R#2 - Discussion

The discussion interprets the findings in the context of existing literature and highlights the implications for public health and policy. The identification of country-specific factors is a valuable contribution.

Authors – We wish to thank the reviewer #2 this positive comment.

R#2 - However, it could benefit from a more critical examination of the study's limitations and potential sources of bias. 

Authors – Thanks for this comment. We have added a new section (i.e. the Section 5.1) including the strengths, limitations and future directions of our study. In particular, we’ve emphasised the potential selection bias (see also next reply below); indeed, we cannot exclude that those who responded the survey may be more interested in the topic discussed than the general population, thus giving a partial picture of the problem. This issue was addressed in the discussions (see Section 5.1 “Strengths, limitations and future directions”, lines 718-720).

R#2 - Discuss the generalizability of the findings and suggest areas for future research. For instance, integrating insights on how lifestyle behaviors and age-related factors influence MD adherence could enhance the discussion (https://www.mdpi.com/2072-6643/15/23/4892, https://www.mdpi.com/2072-6643/15/23/4892). 

Authors – Given the relatively low variance explained by the models across all countries, novel studies are needed to identify a more comprehensive regional list of predictors including, e.g., variables widely applied such as subjective norms, perceived behavioural control, and other psycho-social variables. The current debate on age-gender-MD adherence relationship among studies suggests the possible direction of future research; literature references have been provided. See the integration to the discussion (see Section 5.1 “Strengths, limitations and future directions”, lines 730-745).

R#2 - Conclusion

The conclusion succinctly summarizes the main findings and their implications. It effectively reinforces the need for tailored interventions to promote MD adherence.

Authors – Thanks a lot for this positive comment.

R#2 - Proposing specific policy recommendations based on the study's findings would be a plus.

Authors – Thanks for this suggestion. We have added two sentences with policy implications (lines 760-762 and 766-769).

Reviewer 3 Report

Comments and Suggestions for Authors

Biggi et al conducted a well designed and performed study on a key target, that of MD.

No plagiarism is detected, only minor language editing is suggested.

The statistical analysis does not rise major concerns. 

Minor comments

1. In Table 1: data should be presented as follows: mean(SD) in one column for every country

Table 2: Explain in detail what  " indicate statistically significant differences between countries" means: a and b

Table 3. No legends available, thus, authors are suggested to add in detail what a and b mean

In the discussion section:

Authors are suggested to consider cultural and recipes (thus, ingredients and mode of cooking) variation to their interpretation of the results. 

Comments on the Quality of English Language

No plagiarism or significant errors have been noted

Author Response

Biggi et al conducted a well designed and performed study on a key target, that of MD.

No plagiarism is detected, only minor language editing is suggested.

The statistical analysis does not rise major concerns.

Authors – The authors want to thank Reviewer#3 for his/her comments and suggestions, which contributed improving the data presentation and interpretation.

Minor comments

R#3 - 1. In Table 1: data should be presented as follows: mean(SD) in one column for every country

Authors – Thanks for this point. The Table (in the current version Table A2) has been modified accordingly (see page 22). For consistency, Table A5 has been modified too according to the suggestion of Reviewer #3.

R#3 - Table 2: Explain in detail what  " indicate statistically significant differences between countries" means: a and b

Authors – Thanks for pointing this out. According to the note in Table 2 the presence of different letters in different cells on the same line means that the reported average values are statistically different. Therefore, the mean equal to 7.17 is not statistically different from 7.21, while it differs from 6.06, 7.62, 7.67. When “a, b” is reported in a cell, as in Table 1, it means that the displayed value is equal to those on the same line reporting a or b and different from those reporting c or d. The absence of letters means that there are no statistical differences. The same case can be shown also reporting for all the cells on the same line the letter “a”. This is a quite common way to show the results of pairwise comparisons (see, e.g., https://doi.org/10.2134/agronj2017.10.0580), and we have added “Means followed by a common letter are not significantly different” in the Notes of Tables 2, A2 and A5, and “Medians followed by a common letter are not significantly different” in the Notes of Tables 1, 2, and 4. 

R#3 - Table 3. No legends available, thus, authors are suggested to add in detail what a and b mean.

Authors – Actually, no letters were indicated for Table 3. Instead, we have added the sentence “Means followed by a common letter are not significantly different” in the Notes of Tables 2, A2 and A5, and “Medians followed by a common letter are not significantly different” in the Notes of Tables 1, 2, and 4 (see the reply to the previous comment).

R#3 - In the discussion section:

Authors are suggested to consider cultural and recipes (thus, ingredients and mode of cooking) variation to their interpretation of the results.

Authors – Thanks a lot for this suggestion. We have added a discussion regarding the perceived cultural barrier of adhering to the MD for the Tunisian and Moroccan respondents. We have also discussed that for these respondents it is likely that having a positive attitude towards the MD is not a sufficient driver for adhering more closely to this dietary behaviour, since the behavior is not perfectly aligned with individuals’ cultural values. We have indicated how this finding might support the recent theoretical discussions about the importance of individuals’ goals in forming the motivation to consider performing a particular behavior (Ajzen & Kruglanski, 2019). Finally, we have suggested that for these countries the recovery or rediscovery of traditional recipes in line with the MD and the gastronomic culture could represent a strategy to bridge this attitude-behavior gap (see Section 5, page 16, lines 575-586).

Round 2

Reviewer 3 Report

Comments and Suggestions for Authors

Accept this version